# Multi-Statistical Approach for the Study of Volatile Compounds of Industrial Spoiled Manzanilla Spanish-Style Table Olive Fermentations

**DOI:** 10.3390/foods10061182

**Published:** 2021-05-24

**Authors:** Antonio Garrido-Fernández, Alfredo Montaño, Amparo Cortés-Delgado, Francisco Rodríguez-Gómez, Francisco Noé Arroyo-López

**Affiliations:** Food Biotechnology Department, Instituto de la Grasa (CSIC), University Pablo de Olavide, Building 46, Crta, Utrera km 1, 41013 Seville, Spain; agarrido@ig.csic.es (A.G.-F.); amontano@ig.csic.es (A.M.); acortes@ig.csic.es (A.C.-D.); frgomez@ig.csic.es (F.R.-G.)

**Keywords:** green Spanish-style olives, spoilage, volatile compound, butyric, putrid, sulfidic, compositional data analysis, statistical analysis

## Abstract

Table olives can suffer different types of spoilage during fermentation. In this work, a multi-statistical approach (standard and compositional data analysis) was used for the study of the volatile organic compounds (VOCs) associated with altered (butyric, sulfidic, and putrid) and non-altered (normal) Manzanilla Spanish-style table olive fermentations. Samples were collected from two industrial fermentation yards in Seville (Spain) in the 2019/2020 season. The VOC profiles of altered (*n* = 4) and non-altered (*n* = 6) samples were obtained by headspace solid-phase microextraction combined with gas chromatography-mass spectrometry (HS-SPME-GC-MS). Ninety-one VOCs were identified and grouped into alcohols (30), esters (21), carbonyl compounds (12), acids (10), terpenes (6), phenols (6), sulfur compounds (2), and others (4). The association of the VOCs with spoilage samples depended on the standard or compositional statistical methodology used. However, butyric spoilage was strongly linked by several techniques to methyl butanoate, ethyl butanoate, and butanoic acid; sulfidic spoilage with 2-propyl-1-pentanol, dimethyl sulfide, methanol, 2-methylbutanal, 2-methyl-2-butenal, ethanol, 2-methyl-3-buten-2-ol, and isopentanol, while putrid was mainly related to D-limonene and 2-pentanol. Our data contribute to a better characterisation of non-zapatera spoiled table olive fermentations and show the convenience of using diverse statistical techniques for a most robust selection of spoilage VOC markers.

## 1. Introduction

The use of olive fruit as fermented food is an old tradition in the Mediterranean basin, which later expanded worldwide [1]. The current total production of table olives is slightly above 3.10^6^ tonnes/year, with the EU (30%) as the leading producer. The most demanded trade preparation is Spanish-style. The process consists of lye treatment, washing, and further lactic acid fermentation, leading to products with a pleasant fruity flavor and appropriate mechanical characteristics for pitting, stuffing, or slicing [2]. Legislation on its sensory properties is vague. The only quality criterion is that fermented olives should have normal colour, flavour, and texture [2]. However, such concepts have never been formally defined.

Flavour is closely connected with the quali-quantitative composition of volatile organic compounds (VOCs). It is considered a quality index of olive products [3]. Headspace solid-phase microextraction (HS-SPME) combined with gas chromatography-mass spectrometry (GC-MS) is currently one of the most popular techniques for the analysis of VOCs in foods. In the case of Spanish-style table olives, this technique was used to study the effect of post-fermentation and packaging stages on the VOCs composition [4]. Besides, the effect of processing steps and inoculation with *Lactiplantibacillus pentosus* (ex *Lactobacillus pentosus*) starter culture on the VOC profile [5] and its modulation by several strains of *L. pentosus, Lactiplantibacillus plantarum* and yeast [6] were reported. However, information on VOCs in spoiled table olives is scarce, with zapatera and butyric spoilages being the most studied [7,8].

Concerning the VOCs quantification, data are usually expressed as percentages of the peaks’ total area (which sum a constant value) or are based on a single internal standard peak. In both cases, the estimations are no longer uncorrelated because of sharing the same denominator [9]; besides, the same absolute difference may not reflect the real relative changes [10]. Such data belong to the simplex space, are usually called compositional, and have specific geometric connotations [11,12]. Chayes [13] showed that some correlations between the compositions’ components must be negative because of the constant sum constraint. Later, several authors advised against using for their analysis the standard multivariate techniques developed for data in the Euclidean space [10,12,14]. Furthermore, specific tools for working in the simplex and some algorithms for their transformations into the Euclidean space (that allow analysing them with the standard multivariate methods) have been proposed [10,12]. The application of this strategy is common in different fields such as ecology [15] and meat science [16,17]. Besides, this has led to more realistic results than the direct application of the standard multivariate techniques (principal component analysis or hierarchical cluster analyses) to table olive VOCs [18] or more reliable association between starter cultures with potential markers [19].

This work aims to identify the VOCs related to the non-zapatera table olive spoilage, such as butyric, sulfidic, and putrid, which could be used as potential spoilage markers. For this purpose, a multi-statistical approach, based on both standard statistical techniques and CoDa strategies, was used.

## 2. Materials and Methods

### 2.1. Olives and Sampling

A total of 10 industrial fermentation vessels (10,000 kg olives plus 6000 L brine capacity) were sampled from two companies (A and B) in the Seville province (Spain) during the 2019/2020 season. The olives (Manzanilla cultivar) were prepared according to the Spanish-style method. Samples (*n* = 6) in company A had the following codes: F282, F283, F284, F285, F286, and F309. In company B, the codes of samples (*n* = 4) were F502, F508, F562 and F592. All samples, except two of them (F282 and F283), had been processed following the habitual processing, that is, treatment with a lye solution (2–3% NaOH) for 6–7 h, washing for 6 h, and brining using an acidified brine (8–12% NaCl, 15 L of 37% HCl per vessel). In the case of F282 and F283, the acidified brine contained 2% NaCl instead of 8–12% NaCl. Brine samples from each vessel were taken after one month of fermentation for chemical analysis in our laboratories. The organoleptic evaluation of samples was provided by the technical staffs of the respective companies.

### 2.2. Physic-Chemical Analysis

The analysis of pH and sodium chloride in brine was performed using the routine methods described by Garrido-Fernández et al. [1].

### 2.3. VOC Analysis

Analysis of VOCs was performed by headspace solid-phase microextraction (HS-SPME) combined with gas chromatography-mass spectrometry (GC-MS). An aliquot of brine (5 mL) was inserted into a 15 mL glass vial, and 50 µL of internal standard (5-nonanol, 2 mg L^−1^) was added. The vial was closed and placed in a water bath adjusted to 40 °C. The vial was equilibrated for 15 min at 40 °C and stirred at 600 rpm using a stirring bar. The volatile compounds were extracted for 30 min on a divinylbenzene/carboxen/polydimethylsiloxane (DVB/CAR/PDMS) fibre (2 cm, 50/30 µm; Supelco, Bellefonte, PA, USA). The VOCs adsorbed on the SPME fibre were desorbed at 265 °C for 15 min in the injector port of a GC interfaced with an MS (internal ionisation source: 70 eV) with a scan range from m/z 30 to 400 (7890A-5975C GC-MS system, Agilent Technologies, Santa Clara, CA, USA). The separation was achieved on a VF-WAX MS capillary column (30 m × 0.25 mm × 0.25 μm film thickness) from Agilent. The initial oven temperature was 40 °C (5 min), then 40–195 °C at 3 °C min^−1^, and then 195–240 °C at 10 °C min^−1^, held for 15 min. The carrier gas was helium at a constant flow of 1 mL min^−1^. MassHunter software version B.09.00 (Agilent Technologies, Santa Clara, CA, USA) was used to detect and quantify peaks based on areas as determined by the deconvolution algorithm. A search of the NIST 17 MS library was utilised for the tentative identification of deconvoluted peaks. Chemical names were assigned to peaks with a minimum mass spectral similarity >80 (100 is an exact match). Confirmation was conducted by comparing the retention indices with literature data reported for equivalent columns and real standards. The VOCs were semi-quantified by comparison of peak areas to that of internal standard (5-nonanol).

### 2.4. Statistical Analysis

A multi-statistical approach (standard and CoDa analysis) was used to analyse the data set. The standard evaluation of the VOC profiles (average, standard deviation, ANOVA and classification tree) was performed using XLSTAT v. 2017 for Excel (Addinsoft, Paris, France). For exploratory tools in the simplex (biplot, variation array, and *clr* transformations), the package CodaPack v 1.03 (Department of Computer Sciences and Applied Mathematics, Universidad de Girona, Girona, Spain) ([20] was used. Clustering of samples, complete log-ratio analysis, and selecting the most influential VOC log-ratios was performed using WARD, LRA, and STEP from easyCODA [21]. The detection of potential biomarkers [22] was achieved in robCompositions [23]. The last two packages were run in R (v 4.0.3) (R Foundation for Statistical Computing: Vienna, Austria [24], under RStudio (v 1.4.1103) (RStudio, Inc.: Boston, MA, USA) [25]. Because of the number of cases, PLS-R analysis was not intended.

For CoDa exploratory tools, the original data (in the simplex) were transformed into values in the real (Euclidean) space. The centred log-ratio transformation (*clr*), which preserves distances, is one of the most straightforward procedures [12]. It transforms the composition X into coefficients Y by dividing each cell’s value by the geometric mean of the VOCs in the corresponding row and taking then the logarithm of the resulting ratio. Its formula is:(1)$$Y=\mathrm{clr}(X=[y1,\;y2,\;\ldots .\mathrm{yD}]=ln\frac{x_{1}}{\sqrt[{D}]{\sum_{i=1}^{D}x_{i}}}\;\ldots ln\frac{x_{D}}{\sqrt[{D}]{\sum_{i=1}^{D}x_{i}}},$$

When Y’s variance is estimated column-wise, each VOCs *clr* variance is obtained following the standard procedure. Its magnitude represents the variability of the components over observations. It may be used to measure the effect of factors without omitting the possible influence of parts with low contents and high estimation errors. The *clr* transformation is used to build the compositional biplot [26] and allows a straightforward interpretation of the result when applying the standard multivariate statistics.

For biomarker identification, the statistic Vj developed by Walach et al. [22] was used:(2)$$V_{j}=\sum_{k=1}^{D}\frac{\left(n_{i}+n_{2}\right)\sqrt{\frac{t}{k}}}{n_{1}\cdot \sqrt{t_{jk}^{\left[1\right]}}+n_{2}\cdot \sqrt{t_{jk}^{\left[2\right]}}}\;\mathrm{for}\;j,k=1,\;2,\;\ldots D,$$
where n1 and n2 are the numbers of observations corresponding to the two groups. If the jth part (compound) is a marker, the jth column and row of all the three sources of information will not have a similar structure, and Vj will be markedly higher than 1. Otherwise, the value will be ~1. Vj average and the empirical standard deviations are respectively:(3)$$\bar{V}=\frac{1}{D}\sum_{k=1}^{D}V_{k}\;;\;s_{V}=\sqrt{\frac{1}{D}\sum_{k=1}^{D}\left(V_{k}-\bar{V}\right)},$$

Then, the Vj value can be standardised as $V_{j}^{*}$ = $\frac{V_{j}-\bar{V}}{s_{V}}\;$ and a standard normal percentile for *p* ≥ 0.975 (≈ 1.96) cut off value established. Parts with higher $V_{j}^{*}$ could be considered as potential biomarkers.

## 3. Results and Discussion

### 3.1. Physic-Chemical and Organoleptic Analysis

After one-month fermentation, four samples obtained from industries were considered spoiled. The company’s technical staff defined the spoilages as butyric (F282 and F283), sulfidic (F508), and putrid (F562). The rest of the samples followed the habitual lactic acid fermentation. Butyric deterioration is reminiscent of the smell of rancid butter, the odour of H_2_S gas characterised sulfidic spoilage of olives (the odour is reminiscent of the smell of rotten eggs), and the smell of putrid spoilage is similar to decomposing organic matter [1,8].

All the samples affected by spoilage were among the highest pH values (Figure 1A). Regarding the salt concentrations, most of the normal fermentation vessels have somewhat high levels (particularly those from industry B, except F502), while those affected by butyric spoilage were lower (Figure 1B). An excessively high NaCl level during fermentation could prevent initial alterations, but on the contrary, may inhibit lactobacilli growth. Indeed, NaCl concentrations in F508, F562, and F592 (industry B) may represent an obstacle for proper lactobacilli growth [27]. The fermentation period was still far from reaching a pH equilibrium, but the trends observed indicate that spoiled fermentations vessels from industry A and B will hardly get adequate pH levels for post-fermentation storage, which under normal conditions should be close to 4.0 [1]. Appropriated pH and salt levels are necessary to prevent zapatera and butyric table olive spoilages [8].

### 3.2. Concentration of VOCs in Brine 

The 91 identified and semi-quantified VOC components (Appendix A) were grouped as acids (10), alcohols (30), carbonyl compounds (12), esters (21), phenols (6), sulfur compounds (2), terpenes (6), and others (4). On average, the predominant VOCs were butanoic acid (V62), acetic acid (V50), ethanol (V10), and (Z)-3-hexen-1-ol (V47) (Appendix A). The relatively high contents of butanoic acid (V62) in the samples F282 and F283 can be highlighted, affected by the butyric spoilage. Moreover, sample F282 presented high contents of propanoic (V58) and pentanoic (V68) acids. De Castro et al. [8] investigated the relationships between microbial communities, metabolites, and sensory spoilage descriptors. These authors found that butyric descriptors exhibited a significant positive relationship with genus *Ruminococcus*, which almost significantly correlated with propionic and butyric acids. Levin and Vaughn [28] associated the halophilic *Desulfovibrio aestuarii* with the hydrogen sulfide fermentation of Sicilian-type olives involving the Sevillano variety.

The data set suggests significant differences among samples for many components. However, they do not reflect relative changes, which depend on their total amounts, highly variable in this work (between 50 and 410 ppb, expressed as 5-nonanol) (Appendix A). Then, the comparison among samples in the original values is unreliable. Moreover, the spoilages are perceived when their characteristic VOCs are proportionally higher than in normal samples. Because of such characteristics, these VOCs profiles are compositional data, and the interest of the analysis might be focused on the ratios between components, which require appropriate statistical tools [11].

### 3.3. Relating Samples to VOCs by Univariate Analysis of Variance (ANOVA)

ANOVA is a first, straightforward approach to relate spoiled or normal samples with VOCs. Because of the data structure, the analysis was performed after their *clr* transformation and constraining the sum of model coefficients to 0 (null). The significant positive standardised coefficient for a sample means a relevant contribution to the model (high concentration of the compound in the sample). In contrast, low negative contributions correspond to low presence/absence [6], and their interpretations (out of the scope of the work) will not be intended. In this line, the contribution of F508, affected by sulfidic spoilage, to the response (level of dimethyl sulfide (V2)) is outstanding (Figure 2A) as well as that of F282 and F283 to methyl butanoate (V13) (Figure 2B). Therefore, dimethyl sulfide and methyl butanoate should be considered as, at least, one of the components that characterise sulfidic (F508) and butyric (F282 and F283) spoilages, respectively. The low (but significant) contribution of F562 to the dimethyl sulfide (V2) and F502 to methyl butanoate (V13) models could be interpreted as the result of an overcome initial spoilage.

Including all the ANOVA graphs in the text is not feasible. Instead, the coefficients corresponding to the samples with significant contributions to the different VOCs’ models are summarised in Appendix A. Therein, the most relevant values have been highlighted in bold. The following comments will be focused on them.

#### 3.3.1. Identification of VOCs Associated with a Single Spoilage

The VOCs which could characterise the butyric spoilage (high contribution of both F282 and F283 to their VOCs’ models) were (see Appendix A): methyl butanoate (V13), ethyl butanoate (V18), butanoic acid (V62), and methyl salicylate (V69). Besides, several VOCs were linked to only one of the butyric samples. Methyl propanoate (V8), pentanoic acid (V68), and octanoic acid (V85) were related to F282 (see Appendix A). Conversely, propanoic acid (V58) was related to F282 (mainly) and F283, but the simultaneous contribution of other fermentation vessels from industry A prevents its univocal association with the spoilage.

Regarding sulfidic spoilage, F508 had significant-highly-relevant ANOVA coefficients for the following VOCs: dimethyl sulfide (V2), methanol (V6), 2-methylbutanal (V9), Ethanol (V10), 2-methyl-2-butenal (V24), 1-pentanol (V37), 1-hexanol (V45), and 2-propyl-1-pentanol (V53) (see Appendix A). Then, they characterise this alteration.

F562 sample, affected by putrid spoilage, significantly contributed to the models of D-limonene (V32), 4-ethylphenol (V88), and 2,3-dihydrobenzofuran (V91). The reduced number of VOCs related to this alteration could be due to its incipient stage of development.

#### 3.3.2. VOCs Common to Several Spoilages/Normal Fermentation

The models of some VOCs received significant-high contributions from several spoiled samples. In this line, F282 and F283 (affected by butyric spoilage) and F508 (sulfidic spoilage) contributed to the models for 1-butanol (V30), styrene (V36), and benzoic acid (V91), indicating that they could be produced in both spoilages.

Most of the compounds found in only one sample with butyric fermentation (F2082 or F283) were also related to sulfidic (F508), although in marked lower values. They were butyl acetate (V21), methyl pentanoate (V23), propyl butanoate (V27), ethyl pentanoate (V29), butyl butanoate (V34), (Z)-3-hexenyl acetate (V40), (E)-3-hexen-1-ol (V46), and 3,4-dimethylbenzaldehyde (V71). Thus, they could be related to both spoilages.

Sulfidic (F508) and putrid (F562) spoilages significantly contributed to the following compounds’ models: ethanol (V10), 2-methyl-3-buten-2-ol (V19), isopentanol (V33) and, at lower level, to acetone (V3), methyl acetate (V4), 2-pentanone (V11), and 2-pentanol (V28). They could then be associated with both spoilages.

Also, prenol (V42), (Z)-3-hexen-1-ol (V47), dimethyl sulfoxide (V60), methyl hydrocinnamate (V73), and 3-hydroxy-2,4,4-trimethylpentyl 2-methylpropanoate (V77) could be associated with normal fermentation because of their significant-moderate standardised coefficient values in at least eight out of the 10 sample VOCs’ models. Lastly, hexanal (V22), prenol (V42), (E)-linalool oxide (V52), propanoic acid V(58), o-guaiacol (V76), or p-creosol (V81) were present in at least 5 out of the 6 samples from industry A. However, no VOCs could be related exclusively to industry B, probably because of its high variability in processing conditions. Overall, the ANOVA was generous in associating VOCs with spoilages and habitual fermentations, but could not derive a threshold for establishing an unambiguous relationship between components and type of sample.

Four sensory descriptors (lactic, lupin, wine, and alcohol) were related to VOCs in normal Spanish-style table olives using partial least squares (PLS) regression [29]. The correlation was as follows: lactic odour with methyl propanoate, methyl hydrocinnamate, methyl (E)-3-hexenoate, 1-propanol, and propanoic acid; lupin descriptor with ethyl propanoate, methyl (E)-3-hexenoate, acetic acid, methyl propanoate, and 3-methyl-2-buten-1-ol; wine descriptor with methyl hexanoate, 3-methyl-3-buten-1-ol, geraniol, benzyl alcohol, and β-citronellol; and the alcohol attribute with geraniol, 1-hexanol, methyl hexanoate, ethanol, and benzyl alcohol.

### 3.4. Sample Segregation by VOCs

Classification tree is a potential power tool (data mining) for investigating multilevel interactions [30,31] and classify samples. The technique was used to identify a few VOCs able to segregate the spoiled samples from those following normal fermentation from industries A and B. The rules (Table 1) were based on the values of the *clr coefficients* corresponding to methyl 2-methylbutenal (V24) and acetone (V3) and can predict further classification. The levels of the first VOC, segregated, on the left, the normal samples from industry A with low *clr* values, ≤−4.420. The other two groups, on the center, included butyric brines (company A) and normal samples from industry B, with an intermediate *clr* in the interval (−4.420, −4.016) for methyl 2-methylbutenal (V24); and on the right, sulfidic and putrid olive samples, without segregation between them, with high *clr* levels >−4.016) (Figure 3). After that, the samples with intermediate concentrations (−4.420, −4.016) were re-assigned to their corresponding groups: butyric spoiled (industry A) (cut-off limits ≤−0.365) and normal fermentation (industry B) (Figure 3), based on the *clr* values of acetone. The overall success using the rules mentioned above (Table 2) was 90%, with 100% in most samples (except those affected by sulfidic spoilage, considered as putrid). This misclassification agrees with the ANOVA results because of both types of spoilages’ simultaneous contribution to several VOCs. The classification rules are simple and show that a few compounds could be enough for differentiating spoiled and normal olives, but the technique is not exhaustive.

### 3.5. Relating Samples to VOCs by CoDa Exploratory Analysis

#### 3.5.1. Variation Array and Biplot

Usually, the evolution of components is measured by the central tendency and variance. However, in the compositional analysis, the mean and variance are meaningless. Instead, the analysis is focused on the log-ratios [11]. The variability of the VOCs is estimated, column-wise, as the log-ratio variance of every component over each other. High variances are associated with differences between spoiled and normal fermentation samples. Their values are presented in the so-called variation array (Appendix A), which includes log-ratio variances (upper half) and log-ratio means (bottom half). The highest variance values are observed for methyl butanoate (V13), ethyl butanoate (V18), propanoic acid (V58), or butanoic acid (V62) with different levels according to the VOC in the denominator. However, when the original data are *clr* transformed (coefficients), the individual variances (within, between, or regardless of treatment) of each *clr* transformed VOC (*clr coefficients*) are estimated by the standard procedure. It is named the *clr* variance and represents the *clr coefficients’* variability over treatments (Appendix A, last column at the right). The VOCs more affected by the spoilage (Appendix A) will show the highest values. Among them (Appendix A): ethyl butanoate (V18) (related to butyric spoilage), butanoic acid (V62) (butyric), methyl butanoate (V13) (butyric), 2-pentanone (V11) (sulfidic and putrid), and pentanoic acid (V68) (butyric). The tetrahedral plot of the samples as a function of the first four variables shows differences in contents between F282 and F283 (butyric samples, industry A) and F309 and F285 (normal process, industry A) or between F202 (industry B) and most of the other samples (regardless of industry), which are grouped in the c(V55) vertex. The close position of samples (regardless of industry) means that they have similar VOCs levels (Figure 4). Therefore, the plot was helpful for visualising the differences between samples, although the geometry only allowed for comparison as a function of four components.

The CoDa biplot is one of the most efficient exploratory tools in CoDa analysis [26]. The PC1, PC2, and PC3 explained 42.36%, 18.77%, and 10.36% of the total variance (Figure 5). The biplots showed numerous variables following similar trends (redundant information). Besides, many of them present short arrows, which indicates a poor representation on the PC1 vs PC2 plane (Figure 5A) or PC1 vs PC3 (Figure 5B). Notably, on the PC1 vs PC2 plane, spoiled butyric samples were situated on the left, normal fermentation samples from industry A on the fourth quadrant, and all samples from industry B on the first (Figure 5A). The segregation of butyric samples from the rest was based on log-ratios between compounds on the left divided by those on the right, or vice versa, (e.g., ln[methyl butanoate (V13)/2-pentanol (V28)] or ln[ethanol (V10)/(Z)-3-hexen-1-ol (V48)] which, in turn, were associated to PC1 (Figure 5A). The plot also shows that the highest variance (distance between arrows) also corresponded to the same log-ratios. However, log-ratios along PC2 were considerably lower (e.g., ln[1,4-dimethoxybenzene (V67)/ethyl lactate (V44)], although enough to separate samples from industry A and B and even the two samples affected by the butyric spoilage. Their variances are also low (short distance between arrows). In the presentation on PC1 vs PC3 (Figure 5B), the log-ratio (benzaldehyde (V56)/1-propanol (V17)), with high projections on PC3, is responsible for the proportion of variance explained by this axis and the particular changes observed in the position of several samples (e.g., F283 or 284). Overall, the CoDa biplot showed that the VOC profiles of butyric samples differed from the rest, based on PC1, and segregated samples from industry A and B (based on PC2/PC3). Besides, it allows identifying the log-ratios responsible for the segregation and their components. However, as an exploratory technique, the relationships mentioned above should be considered exclusively qualitative.

#### 3.5.2. Relating Samples and VOCs by Clustering

The dissimilarity between the VOC profile of spoiled and normal samples was also studied by clustering, a technique successfully used for segregating according to starter culture [6]. The technique was applied to the *clr* transformed data set (*clr coefficients*), using the function WARD from easyCoDa with the unweighted and weighted options (Figure 6). The VOC profiles of the butyric spoiled brines (F282 and F283) were always consistently recognised as singular and different from any other sample. The association agrees with that observed in the CoDa biplot (Figure 5). Besides, regardless of the option, putrid (F562) and normal (F592) samples from industry B were also considered alike, possibly because of the early initial stage of the spoilage in the first. The grouping agrees with the ANOVA results where F562 had a limited contribution to most VOCs models. However, the segregation between sulfidic and normal/putrid samples from industry B and grouping normal samples depended on the cluster option. In the unweighted cluster (Figure 6A), F562, F592, and F508 (sulfidic) from industry B (F562, F592, and F508) were grouped, although considering sulfidic sample (F508) somewhat different; on the contrary, in the weighted option (Figure 6B), segregation of sulfidic sample from the other two samples from industry A was straightforward and more reasonable. In addition, clustering normal processes in the unweighted option grouped those from industry A (including F502, from industry B) separately from industry B. In the weighted dendrogram, they were included together and considered relatively close.

The same easyCODA function was applied for detecting groups of VOCs (Q-Q clustering) which could be related to specific spoilages. The difference between the unweighted and weighted options was marked (Figure 7), showing the first a greater degree of dissimilarity and more challenging interpretation. Both options agree to segregate cluster C1 on the left composed of butanoic acid (V62), ethyl butanoate (V18), and methyl butanoate (V13), significantly related to the two samples (F282 and 283) affected by the butyric spoilage (Appendix A).

Regarding the unweighted option, C2 cluster included highly significant and almost exclusive compounds associated with the ANOVA to sample F282 (affected by the butyric spoilage) (Appendix A). Similarly, most of the VOCs grouped into cluster C7 were characterised by the exclusive significant-high contribution of F508 (affected by sulfidic spoilage) to their ANOVA models. Besides, (Z)-3-hexen-1-ol (V47) was present in all the samples and dimethyl sulfoxide in all except in F282 and F283. Among the compounds included in cluster C8, only D-limonene (V32) was exclusively related to F562. The rest of the components were mainly linked by the ANOVA to sample F562 (putrid) and other samples. VOCs included in cluster C3 were primarily associated with samples from industry A, including F282 and F283. C4 was composed mainly of VOCs from samples F282 and F283 (putrid) or industry B like F508 and F562, affected by sulfidic and putrid. C6 grouped compounds related to most samples, while VOCs included in C5 did not follow identifiable trends.

Concerning the weighted option (Figure 7B), apart from the three VOCs in C1, other associations with the ANOVA results were not straightforward. C2 mainly included VOCs linked by the ANOVA to F282 but only methyl propanoate (V8), pentanoic acid (V68), and octanoic acid (V85) in exclusive to this sample. Furthermore, several were also linked to F508 and others to the normal fermentation from industry A. C3 was a complex group hardly associated with any spoilage. C4 was formed by only (Z)-3-hexen-1-ol (V47) related to most samples (except F282). Finally, C5 consisted mainly of F562 and F592 (industry B) VOCs and several from industry B.

Overall, the unweighted clustering agrees with the ANOVA models in grouping the VOCs related to spoilages and could represent a complementary tool for associating them with specific spoilages. In this case, the unweighted option, apparently, better represents the reality.

### 3.6. Association of Most Influential VOCs to Spoilage

A procedure for identifying a set of influential log-ratios while simultaneously controlling the progressive variance explained by the selection was achieved using the function STEP of the easyCODA package. After six steps, the corresponding log-ratios accounted for 97.87% of the variance (Table 3). The VOCs involved in such a log-ratios were: methyl propanoate (V8), 2-pentanone (V11), 2-methyl-2-butenal (V24), (Z)-3-hexenyl acetate (V40), (Z)-2-hexen-1-ol (V47), (E)-2-hexen-1-ol (V48), 5-hexen-1-ol (V49), butanoic acid (V62), β-damascenone (V72), ethyl hydrocinnamate (V79), and 4-ethyl phenol (V88). However, notice that the selection is not unique, and other sets of log-ratios could also have been chosen.

As such log-ratios are data in the Euclidean space, they can be subjected to the usual PCA analysis (Figure 8A), which separated the samples affected of butyric spoilage, based on log-ratio ln[(Z)-3-hexen-1-ol (V48)/butanoic acid (V62)], strongly and negatively related to PC1; a low value of this log-ratio characterises then the butyric samples. Besides, two samples from industry B (F508, with sulfidic spoilage, and F502, less strongly) were associated with ln(2-pentanone (V11)/β-damascenone (V72)), which was negatively linked to PC2, but the rest of samples were considered somewhat similar. The PCA based on these log-ratios did not distinguish between normal fermentation in industry A and the putrid (F562)/normal (F592) samples from B, as already shown by other analyses like clustering, but identified several log-ratios that accounted for most of the original dataset variance.

The plot using this reduced number of VOC log-ratios is, theoretically, similar to that obtained by the function LRA from the easCODA package (Figure 8B). They were pretty similar after rotation, except for F502 (industry B), not segregated. The advantage of Figure 8B is that it allows associating the leading individual VOCs’ contributors with samples. The VOCs with the highest contribution to variability were: butanoic acid (V62), ethyl butanoate (V18), and methyl butanoate (V13) related to butyric spoilage. This contribution agrees with that found in the ANOVA analysis and Q-Q clustering. Besides, ethanol (V10) and isopentanol (V33) were related to sulfidic spoilage, which again agrees with the high contribution of F508 to the ANOVA models of these VOCs. Besides, (Z)-3-hexen-1-ol (V47), acetoin (V38), and acetic acid (V50) seem to be present in most of the rest of the samples but absent/low in those affected by the butyric and sulfidic spoilages. Notice that several of the influential VOCs were also included in the selected log-ratios.

Therefore, these analyses allowed identifying log-ratios and VOCs directly related to butyric and sulfidic spoilages but failed to detect those associated with putrid. The biplot based on the VOCs constituting these log-ratios also segregated butyric (F282 and F283) and sulfidic samples (F508) and showed similar VOC profiles of the rest of the samples (F284, F285, F286, and F309 from industry A, and F502, F562, and F592 from industry B), similarly to the grouping observed in the weighted clustering option. Therefore, the selection of high contributors’ log-ratios and VOCs to the variance represents a valuable technique to visualize the relationship between them and the samples, while focusing on only the most influential ones.

### 3.7. Identification of Putative VOC Markers

The characteristic VOCs of spoiled samples vs the normal fermentations were analysed using the Walach et al. [22] test. It consisted of comparing the pairwise log-ratio variation array matrix corresponding to the entire data set with those estimated considering independently spoiled and normal fermentation samples. According to Equation (2), $V_{j}$ will then be considerably higher for markers than for non-markers. Thus, the higher the value of this statistic is, the less similar the groups are regarding this j variable. When the results are expressed in terms of $V_{j}^{*}\;\left(\mathrm{Equation}\;\left(3\right)\right),$ values of VOCs exceeding the 1.96 cut-off limit (*p* < 0.05) are considered significant and potential markers. The interest was focused on testing the spoiled fermentation vessels against those following normal processes in the same industry.

#### 3.7.1. Butyric vs Normal Samples from Industry A

In industry A, the comparison was F282 and F283 (butyric) vs F284, F285, F286, and F309 (normal fermentation). The following VOCs had normalised $V_{j}^{*}$ above the 1.96 cut-off limit: methyl butanoate (V13), ethyl butanoate (V18), and butanoic acid (V62) (Figure 9A). Then, they showed significant high dissimilarity (high content) between the spoiled and normal samples. 

One can notice the identification of butanoic acid, having a sharp, cheesy, and dairy-like odor (Perflavory database, www.perflavory.com accessed on 19 April 2021), and its methyl and ethyl esters (fruity notes) in the butyric samples. They could be considered distinctive of this spoilage, as also recognised by other methodologies. On the contrary, the VOCs which characterised the normal process in industry A (low presence in butyric samples vs the others from the same company) were: methyl 2-methylbutanoate (V14), 2-pentanol (V28), 2-ethyl-1-hexanol (V54), and dimethyl sulfoxide (V60).

#### 3.7.2. Sulfidic vs Normal Samples from Industry B

After the comparison of sulfidic spoilage (F508) against the normal fermentation samples in industry B (F502 and F592), there was only one potential biomarker (Figure 9B): 2-propyl-1-pentanol (V53). The odour description of 2-propyl-1-pentanol is not available in the literature. No other compound strongly associated by the ANOVA to this spoilage was so dissimilar (high $V_{j}^{*}$) for being significant or linked to the normal fermentation of this company either.

#### 3.7.3. Putrid vs Normal Fermentation Samples from Industry B

The comparison between the putrid sample from industry B and their normal process (F502 and F592) (Figure 9C) led to identifying as potential markers for this spoilage the following VOCs: 2-methyl-3-buten-2-ol (V19) (herbal, earthy, and oily notes) and ethyl 2-methylbutanoate (V20) (fruity odour), and D-limonene (V32) (sweet, citrus, and peely odour). The absence of malodorous components as biomarkers may again be related to the initial stage of the spoilage and the low presence of VOVs responsible for the unpleasant smell. On the contrary, this comparison pointed out the production of acetaldehyde (V1), ethyl lactate (V44), benzyl acetate (V66), phenylethyl alcohol (V80), 1-dodecanol (V82), and nonanoic acid (V87) by normal samples. Then, one could identify the normal fermentation in industry B.

In general, identifying the potential biomarkers is one of the most selective and restrictive techniques for the association of VOCs with each type of sample, and is therefore quite reliable.

### 3.8. Relationship of Relevant VOCs and Spoilage by Heatmap

The overall linkage of the previously associated with spoilages’ VOCs and the samples was obtained by a heatmap, using their *clr coefficients* (Figure 10).

In industry A, the row dendrogram considered different F282 (R1) and F283 (R2), affected by butyric spoilage, although they have in common several compounds. These were (left to right) butanoic acid (V62), benzoic acid (V91), methyl salicylate (69), ethyl butanoate (V18), methyl butanoate (V13), and styrene (V36). Nevertheless, each of these samples were also associated with specific VOCs. F282 (R1) was characterised by the additional presence of 1,4-dimethoxibenzene (V67), methyl propanoate (V8), (E)-3-hexen-1-ol (V46), pentanoic acid (V68), butyl propanoate (V34), propyl butanoate (V27), ethyl pentanoate (V29), methyl pentanoate (V23), and butyl acetate (V21). F283 (R2) was individually linked to ethyl lactate (V44), benzaldehyde (V56), (Z)-3-hexenyl acetate (V40), and ethyl hydrocinnamate (V79). The other normal fermentations (F284, F285, and F286, F309) were grouped in R7, and their VOC profiles were almost plain in the heatmap (based on compounds selected because of their relationship with spoilages), except for the presence of (E)-2-hexen-1-ol (V48), in F285 and F286, or octanoic acid (V85), in F286 (Figure 10).

Regarding samples from industry B, F508 (R6) was again considered different from the other samples from this company (R8). However, notice the inclusion of F502, F562 (only incipient putrid spoilage), and F592, and the normal samples from industry A in cluster R5, recognising such grouping the common characteristics associated with normal fermentations. F508 (R10), affected by sulfidic spoilage, was characterised in exclusive by 2-propyl-1-pentanol (V53), 2-methyl-2-butenal (V24), 2-methyl-3-buten-2-ol (V19), 2-pentanone (V11), isobutanol (V25), methanol (V6), dimethyl sulfide (V2), ethanol (V10), acetone (V3), isopentanol (V33), and 2-methylbutanal (V9). This association agrees with those from the ANOVA. The F508 (sulfidic spoilage) also shared 1,4 dimethoxibenzene (V67) with F282 (butyric). F502 (R9) was also separated from F562 and F592 (R10) and had a weak relationship with any compound related to spoilage. Its segregation agrees with various of the previous plots (Figure 6A). F562 (affected by initial putrid spoilage) and F592 were included in the same cluster (R10), which were associated with an almost plain VOC profile. However, F562 was particularly linked to 2-pentanol (V28), which could then be related to its detected (initial) putrid spoilage.

Heatmap had the advantage of combining clustering of both cases and variables and then associating groups relatively similar, circumstance that allows a more straightforward comparison of the relationships between samples and VOCs.

### 3.9. Overall Summary of the Association of VOCs with Spoilages

The combined presentation of the associations between spoilages and VOCs by the different analyses would provide an overall picture of their relationships (Appendix A). Notice that the linkage depended upon the analysis and supports the convenience of using diverse statistical methodologies to select putative VOC markers. Thus, only compounds related to each spoilage by several statistical techniques will be finally chosen. The VOCs most strongly associated with butyric (two samples and three techniques) were: methyl butanoate (V13), ethyl butanoate (V18), and butanoic acid (V62). Also related to butyric (two samples and two analysis) were methyl salicylate (V69), styrene (V36), and benzoic acid (V91). There were other compounds related to butyric spoilage by two analysis, but the presence in only one of the samples prevented their selection. The only VOC associated with the sulfidic spoilage by three statistical techniques was 2-propyl-1-pentanol (V53). However, there were also several compounds related to only this spoilage and suggested by 2 methods: dimethyl sulfide (V2), methanol (V6), 2-methylbutanal (V9), 2-methyl-2-butenal (V24), ethanol (V10), 2-mehtyl-3-buten-2-ol (V19), isopentanol (V33), and acetone (V3). Finally, VOCs associated with putrid spoilage by at least 2 statistical techniques were: D-limonene (V32) and 2-pentanol. Besides, 2-Methyl-3-buten-2-ol (V19) and ethyl-2-methylbotanoate (V20) were related to putrid only by the biomarker analysis.

The other VOCs shown in Appendix A were also related to spoilages, but the association was not exclusive, observed in only one analysis, or related by techniques not providing direct linkages.

These results contrast with the main contributors reported by De Castro et al. [8] for butyric spoilage (benzyl pentanoate, butanoic acid, nonanal, pentanoic acid, 1-propanol, dimethyl sulfide, methyl pentanoate, (Z)-3-hexen-1-ol, phenylethyl alcohol, methyl hexanoate, and octanal). Only butanoic acid, pentanoic acid, and methyl pentanoate (found in F282) were associated with the so-called butyric spoilage in this work. Furthermore, most VOCs related to zapatera fermentations by De Castro et al. [8] (propionic acid, methyl propanoate, propyl propanoate, acetic acid, 3-methyl-1-pentanol, 1-octanol, α-terpineol, 1-butanol, and cyclohexanoic acid) were not linked to any alteration in this work, except 1-butanol (identified in F282 and F283) and methyl propanoate (found only in F282), linked to the butyric spoilage. Moreover, cyclohexanoic acid, the typical current marker for zapatería [7], was utterly absent in the studied samples. Therefore, one should deduce from these comparisons that the spoilages analysed in this study were different from those previously investigated, that the definitions of spoilages by industries are not still clear, or that appreciation of the off-flavours usually noticed in table olives can be produced by different combinations of components.

## 4. Conclusions

This work has used a multi-statistical approach to relate non-zapatera alterations with their VOC profiles. The study demonstrates that the selection of spoilage markers may depend on the statistical technique used. This way, our survey represents a relevant contribution to the VOCs that characterise butyric, sulfidic, and putrid table olive spoilages and opens new strategies for further studies on VOC profiles in foods. Because of the diversity of VOCs that each spoilage may present, establishing their definitive profiles requires broader information, including more industries and processing conditions.

## Figures and Tables

**Figure 1 foods-10-01182-f001:**
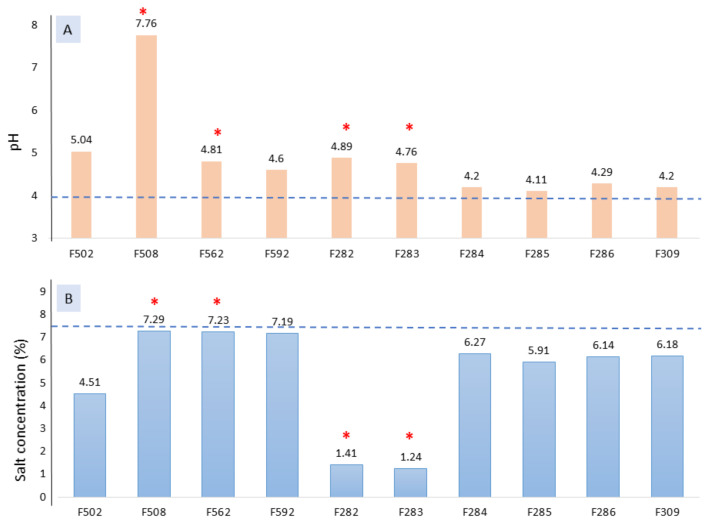
pH (**A**) and salt (**B**) values obtained for the different fermentation vessels at 1 month of fermentation. The horizontal line stands for habitual values under normal conditions for lye treated olive post-fermentation storage [1]. * Spoiled samples.

**Figure 2 foods-10-01182-f002:**
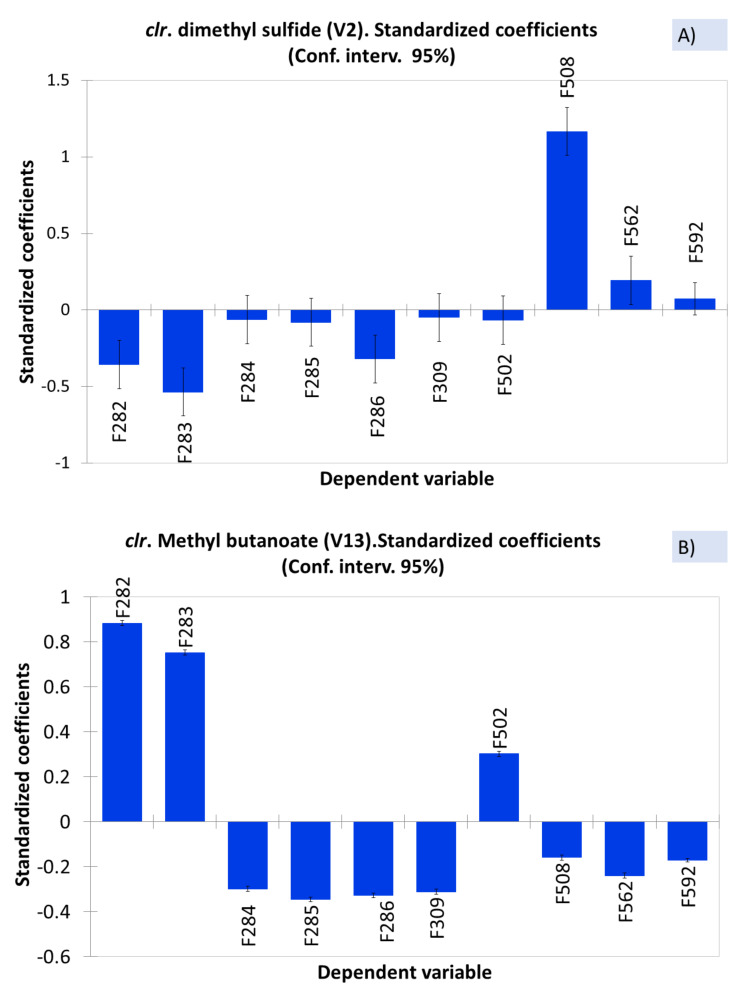
Example of the samples’ contribution to dimethyl sulfide (**A**) and methyl butanoate (**B**) (as *clr coefficients*), based on the standardised ANOVA coefficients. Symbols of samples correspond to industry A (F282 and F283, affected by butyric spoilage; F284, F285, F286, and F309, normal fermentations) and industry B (F508 and F562, affected by sulfidic and putrid spoilages, respectively; F502 and F592, normal fermentations).

**Figure 3 foods-10-01182-f003:**
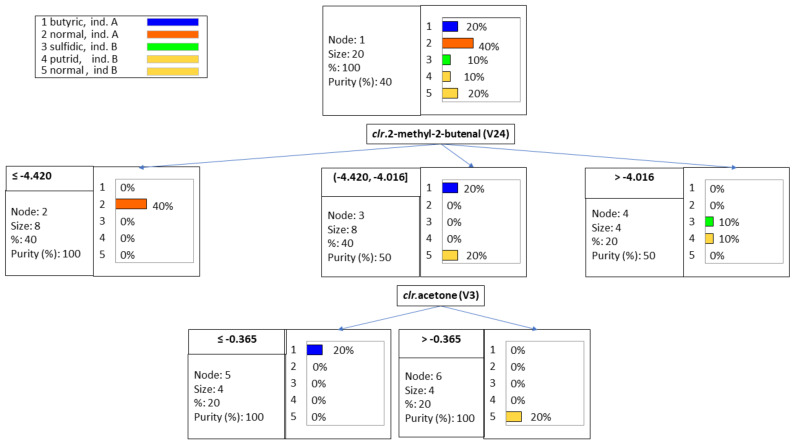
Classification tree of samples, using the *clr coefficients* of the original data set. Notice the simple prediction by just a few components.

**Figure 4 foods-10-01182-f004:**
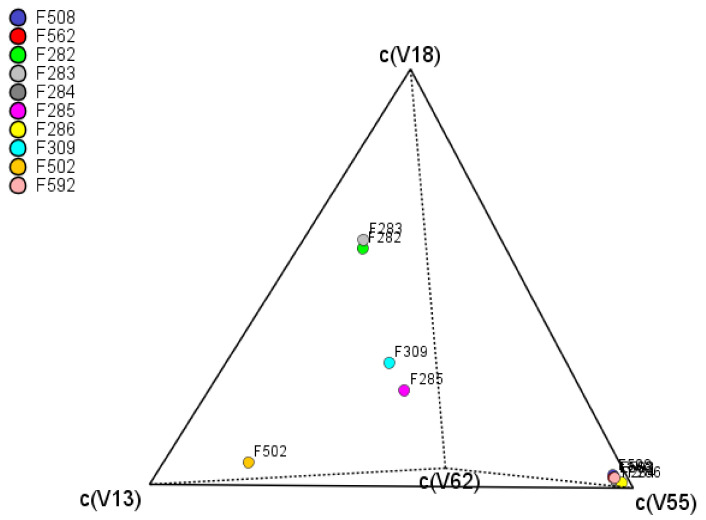
Tetrahedral centered plot of samples as function of the four VOCs with the highest *clr* variance (methyl butanoate (V13), ethyl butanoate (V18), 6-hepten-1-ol (V55), and butanoic acid (V62)).

**Figure 5 foods-10-01182-f005:**
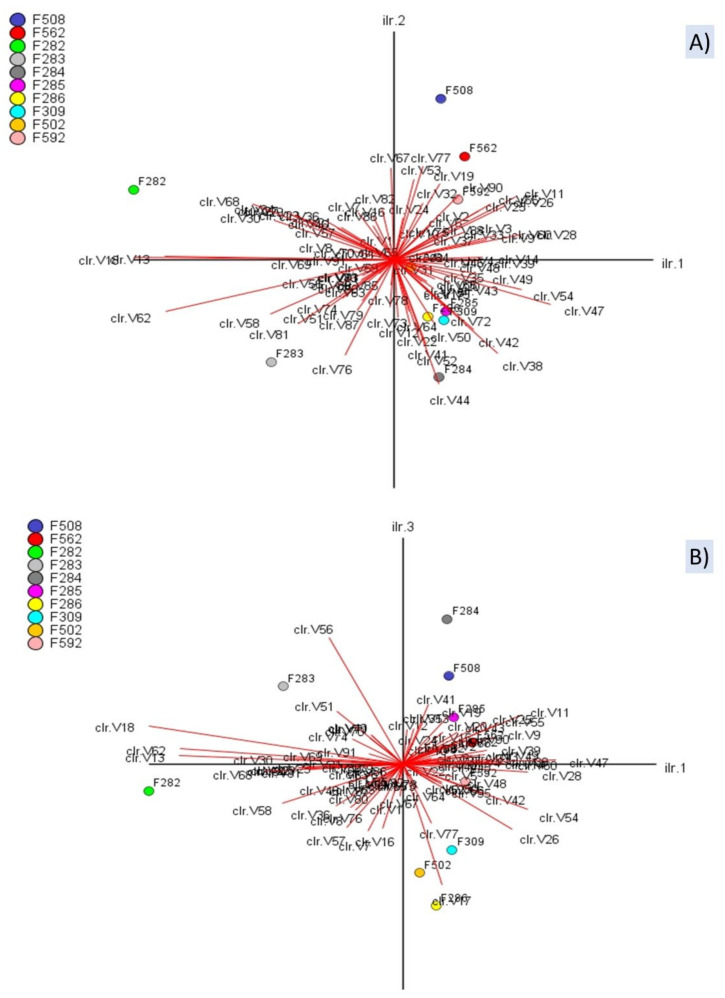
CoDa biplot of the VOCs and samples. Projections on the planes 1st vs 2nd (**A**) and 1st vs 3rd (**B**) dimensions. Symbols of samples correspond to industry A (F282 and F283, affected by butyric; F284, F285, F286, and F309, normal fermentations) and industry B (F508 and F562, sulfidic and putrid spoilage, respectively; F502 and F592, normal fermentations). Symbols for VOCs are detailed in Appendix A. *clr* stands for *clr coefficients*.

**Figure 6 foods-10-01182-f006:**
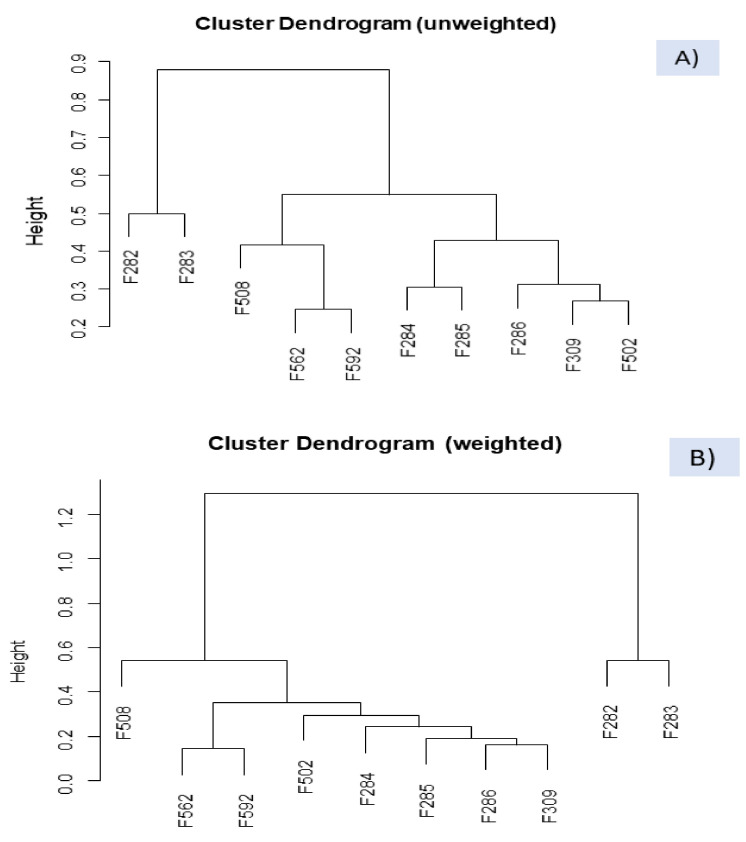
Clustering of samples, based on *clr coefficients*, using the WARD function from easyCODA and the unweighted (**A**) and weighted (**B**) options. Symbols of samples correspond to industry A (F282 and F283, affected by butyric spoilage; F284, F285, F286, F309, normal fermentation) and industry B (F508 and F562, sulfidic and putrid spoilage, respectively; F502 and F592, normal fermentations).

**Figure 7 foods-10-01182-f007:**
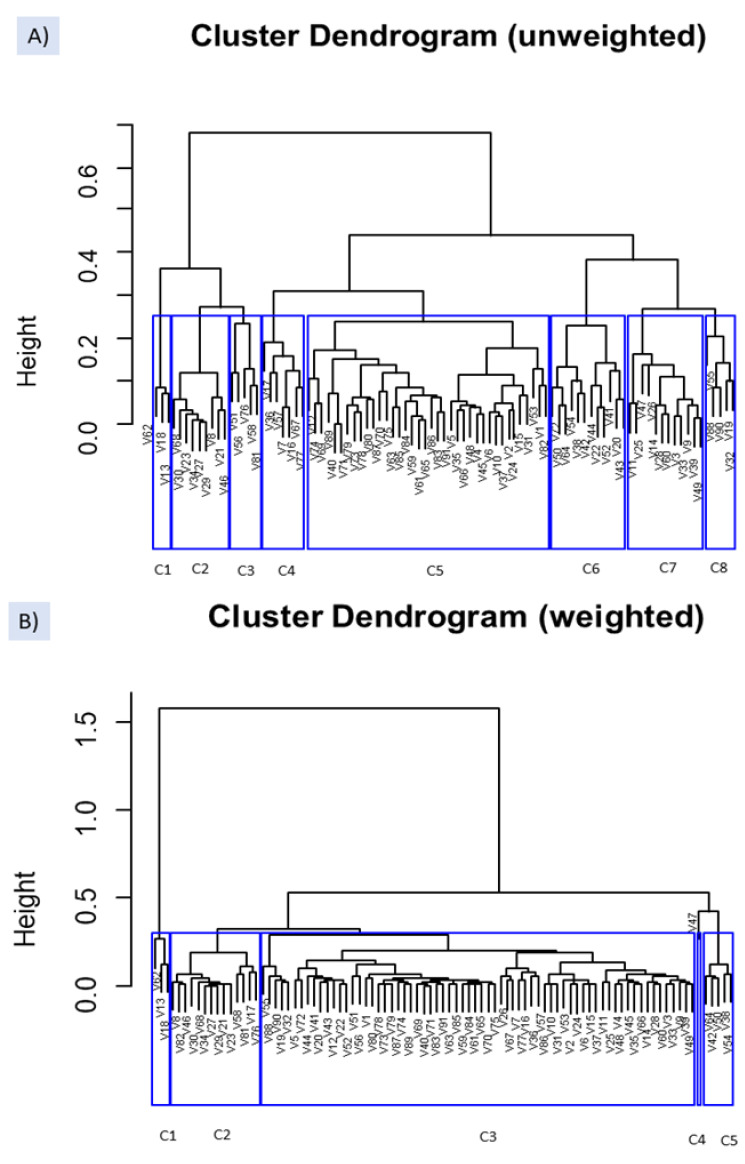
Clustering of VOCs, based on *clr coefficients*, using the function WARD from easyCODA with the unweighted (**A**) and weighted (**B**) options. See Appendix A for the meaning of VOCs’ symbols.

**Figure 8 foods-10-01182-f008:**
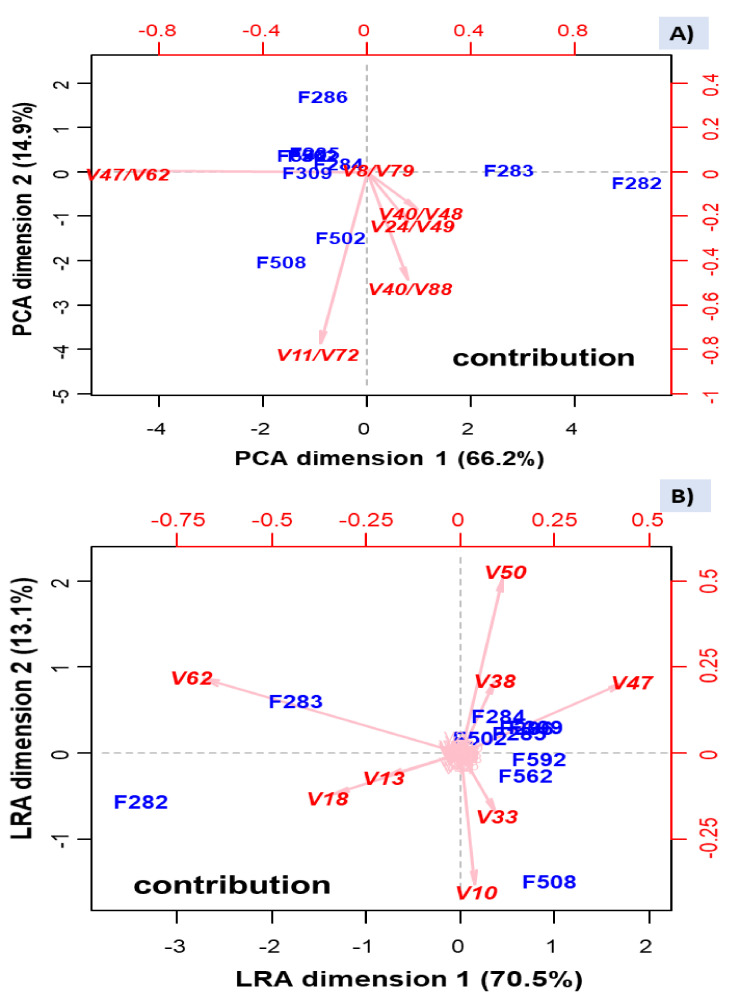
PCA (**A**), based on the log-ratios selected using the function STEP from easyCODA, and log-ratio analysis (LRA) (**B**), based the complete set of log-ratios. Symbols of samples correspond to industry A (F282 and F283, affected by butyric spoilage; F284, F285, F286, F309, normal fermentation) and industry B (F508 and F562, sulfidic and putrid spoilage, respectively; F502 and F592, normal fermentations). See Appendix A for the meaning of VOCs’ symbols.

**Figure 9 foods-10-01182-f009:**
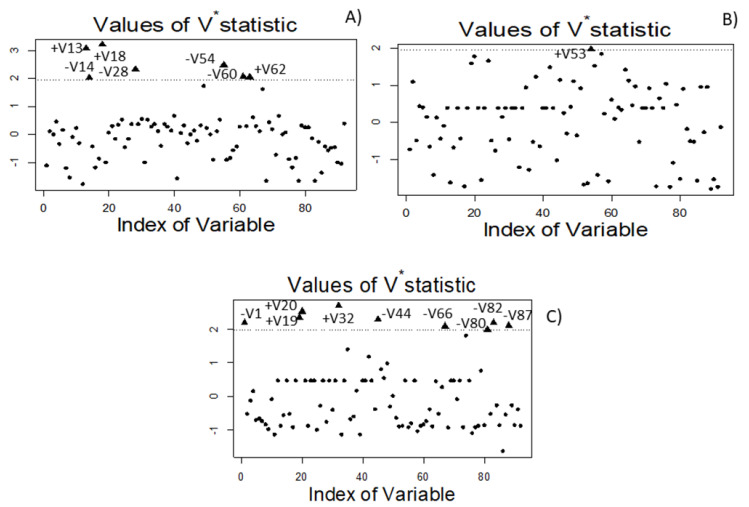
Biomarkers identified using the Walach et al. [22] test. Plots correspond to, (**A**) spoiled butyric samples (F282 and F283) vs normal samples (F284, F285, F286, and F309) from industry A; (**B**) sulfidic (F508) vs normal samples (F502 and F592) from industry B; and (**C**) putrid (F562) vs normal samples (F502 and F592) from industry B. See text for correspondence between VOC indexes and their names. V*statistic stand for the standardized Walach et al. [22].

**Figure 10 foods-10-01182-f010:**
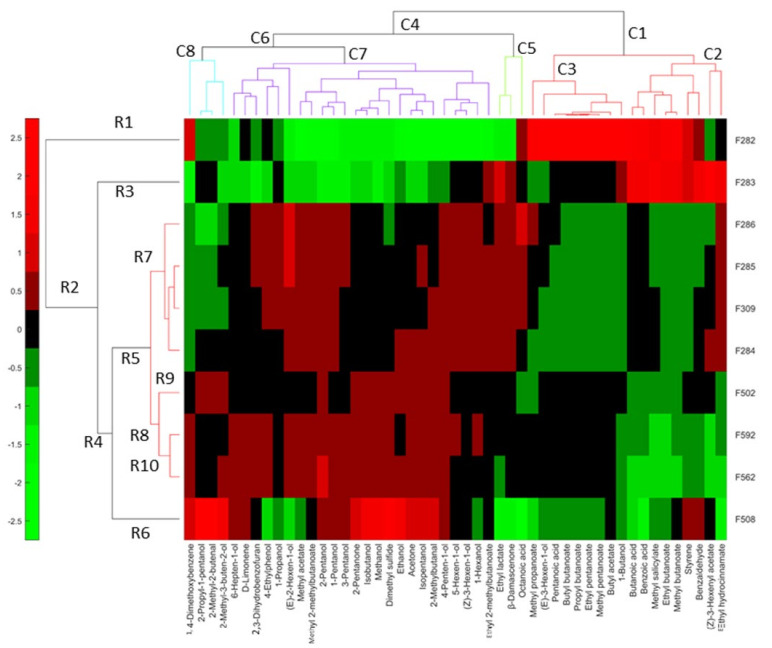
Heatmap based on only those VOCs associated with spoilages by at least one previous statistical technique. Symbols of samples correspond to industry A (F282 and F283, affected by butyric spoilage; F284, F285, F286, F309, normal fermentation) and industry B (F508 and F562, sulfidic and putrid spoilage, respectively; F502 and F592, normal fermentations).

**Table 1 foods-10-01182-t001:** Classification tree analysis. Rules for the classification of spoiled and normal Spanish-style Manzanilla fermentation samples. Codes are as follows: 1, butyric fermentations from industry A; 2, normal fermentations from industry A; 3, sulfidic spoilage from industry B; 4, putrid spoilage from industry B; and 5, normal fermentations from industry B.

Nodes	CODES (Prediction)	Rules
Node 1	2	
Node 2	2	If *clr*.2-methyl-2-butenal (V24) ≤−4.42041 then CODES = 2 in 40% cases
Node 3	5	If *clr*.2-methyl-2-butenal (V24) −4.42041, −4.01629] then CODES = 5 in 40% cases
Node 4	4	If *clr*.2-methyl-2-butenal (V24) >−4.01629 then CODES = 4 in 20% cases
Node 5	1	If *clr*.2-methyl-2-butenal (V24) (−4.42041, −4.01629] and *clr*.acetone (V3) ≤−0.412896 then CODES = 1 in 20% cases
Node 6	5	If *clr*.2-methyl-2-butenal (V24) (−4.42041, −4.01629] and *clr*.acetone (V3) >−0.412896 then CODES = 5 in 20% cases

**Table 2 foods-10-01182-t002:** Classification tree analysis. Predicted confusion matrix of spoiled and normal Spanish-style Manzanilla table olive fermentation samples. The codes are as follows: 1, butyric fermentations from industry A; 2, normal fermentations from industry A; 3, sulfidic spoilage from industry B; 4, putrid spoilage from industry B; and 5, normal fermentations from industry B.

From\To	1	2	3	4	5	Total	% Correct
1	4	0	0	0	0	4	100.0
2	0	8	0	0	0	8	100.0
3	0	0	0	2	0	2	0.0
4	0	0	0	2	0	2	100.0
5	0	0	0	0	4	4	100.0
Total	4	8	0	4	4	20	90.0

**Table 3 foods-10-01182-t003:** Spoiled (*) and normal samples of Spanish-style Manzanilla table olives fermentations. Most influential VOCs’ logratios progressively selected by the function STEP of the easyCODA package, and their cumulative proportions of explained variance.

	Log-Ratios					
	V47/V62	V24/V49	V40/V88	V40/V48	V8/V79	V11/V72
**Sample**	1st step	2nd Step	3rd step	4th step	5th step	6th step
*** F508**	11.5366736	−3.34584729	−0.79449967	−11.5366736	−5.24925185	−3.60413823
*** F562**	4.31194432	−5.26685746	9.25254389	−0.39287304	<10^−8^	2.36124225
*** F282**	<10^−8^	<10^−8^	<10^−8^	10.1927924	4.28393332	11.0951384
*** F283**	<10^−8^	<10^−8^	−4.18599046	10.3222438	−4.18599046	11.0754892
**F284**	8.35119765	8.61352368	−3.35607594	−8.35119765	−10.3377366	11.0976654
**F285**	8.27834396	7.27272682	−0.85347747	−8.27834396	−8.97662228	11.338771
**F286**	8.89742579	6.63844165	−1.35585478	−8.89742579	−10.2055787	11.0516611
**F309**	6.64586886	7.28309525	−1.65695554	−6.64586886	−9.85671251	11.3299799
**F502**	8.36178107	2.52536384	−3.21526325	−0.00764613	−10.8405329	2.59544661
**F592**	5.45442103	−7.2474397	9.22333371	0.09096199	<10^−8^	1.99738057
Cummulative variance (%)	69.35	82.9	90.35	93.86	96.32	97.87

Notes: V8, metyl propanate; V11, 2-pentonone; V24, 2-methyl-2-butenal; V40, (Z)-3-hexenyl acetate; V47, (Z)-3-hexen-1-ol; V48, (E)-2-hexen-1-ol; V49, 5-hexen-1-ol; V62, butanoic acid; V72, β-damascenone; V79, ethyl hydrocinnamate; V88, 4-ethyl phenol. * Spoiled samples.

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
