# Peer review of "Multi-Statistical Approach for the Study of Volatile Compounds of Industrial Spoiled Manzanilla Spanish-Style Table Olive Fermentations"

_foods, 2021, doi:10.3390/foods10061182_

Round 1

Reviewer 1 Report

The purpose of this manuscript was to apply a multi-statistical approach based on univariate data analysis as well as compositional data analysis for the elucidation of the volatile profile of industrially fermented Spanish-style green olives of cv. Manzanilla that were characterized by specific spoilage (i.e., butyric, sulfidic and putrid). I have no special comment to make on this work. The experimental part is correct and the data were analyzed in great detail and depth based on different statistical methods and the conclusions are completely supported by the experimental data and the respective analysis.

My minor comments on this work are the following:

Page 5, lines 15-20 and Figure 1: The values of the pH and salt concentration defined by the trade standard of the IOOC refer to the characteristics of the packing brine or the olive juice after osmotic equilibrium and not to the characteristics of the brine in fermentation vessels, especially when the samples were taken after one month of fermentation where the process was still in progress and had not been completed. Please elucidate.

Page 6, line 37: Please consider replacing “viable” by “feasible”.

Page 32, lines 13-14: In the implementation of the classification tree analysis, please indicate on what grounds two specific volatile compounds, namely 2-methyl-2-butenal and acetone, were selected to predict the classes of olives.

Page 12, lines 3-4: what do the authors mean by “…variables following similar directions provide redundant information”?

Author Response

We sincerely thank to reviewer for his/her constructive comments, which in our opinion, have improved the quality of the revised manuscript.  All changes are been marked in red in the revised version.

Reviewer 1

Q1. Page 5, lines 15-20 and Figure 1: The values of the pH and salt concentration defined by the trade standard of the IOOC refer to the characteristics of the packing brine or the olive juice after osmotic equilibrium and not to the characteristics of the brine in fermentation vessels, especially when the samples were taken after one month of fermentation where the process was still in progress and had not been completed. Please elucidate.

A1. We acknowledge the reviewer comment. The pH limit (the only limit depending on the fermentation) has been accommodated to the levels expected after fermentation and the post fermentation storage.  

Q2. Page 6, line 37: Please consider replacing “viable” by “feasible”

A2. The suggested change was introduced in the revised version.

Q3. Page 32, lines 13-14: In the implementation of the classification tree analysis, please indicate on what grounds two specific volatile compounds, namely 2-methyl-2-butenal and acetone, were selected to predict the classes of olives.

A3. A short sentence to introduce the type do data on which the segregation was based was introduced.

Q4. Page 12, lines 3-4: what do the authors mean by “…variables following similar directions provide redundant information”?

A4. The term “directions” was substituted with “trends” which may catch better the idea behind the expression.

Reviewer 2 Report

The manuscript describes the application of different statistical approaches for the study of the VOC profile. However, the result section is not easy to read; too many details are given making hard to follow the manuscript, i.e. sections 3.3.1. and 3.3.2.

Furthermore, I would suggest adding a critical discussion on pros and cons of each statistical approach.

Figure 1: make it sharper.

Author Response

We sincerely thank to reviewer for his/her constructive comments, which in our opinion, have improved the quality of the revised manuscript.  All changes are been marked in red in the revised version.

Reviewer 2

Q5. The manuscript describes the application of different statistical approaches for the study of the VOC profile. However, the result section is not easy to read; too many details are given making hard to follow the manuscript, i.e. sections 3.3.1. and 3.3.2.

A5. As requested by the reviewer, the text regarding Results and Discussion has been simplified and many details removed from it.

Q6. Furthermore, I would suggest adding a critical discussion on pros and cons of each statistical approach.

A6. As suggested, several sentences have been added to the already existing comments on the line mentioned by the reviewer or have been introduced de novo when absent.

Q7. Figure 1: make it sharper.

A7. According to reviewer’s suggestion, Figure 1 has been improved.

Round 2

Reviewer 2 Report

The authors have address the comments I made in the previous review round.